# Lentivirus-mediated gene therapy for Fabry disease

Aneal Khan[1], Dwayne L. Barber [2,3], Ju Huang[2], C. Anthony Rupar[4,5,6], Jack W. Rip[4], Christiane Auray-Blais[7], Michel Boutin[7], Pamela O'Hoski[8], Kristy Gargulak[9], William M. McKillop[9], Graeme Fraser[10], Syed Wasim[11], Kaye LeMoine[12], Shelly Jelinski[13,14], Ahsan Chaudhry[15], Nicole Prokopishyn[16], Chantal F. Morel[17], Stephen Couban[18,24], Peter R. Duggan [19], Daniel H. Fowler[20], Armand Keating[2,21], Michael L. West[22], Ronan Foley[8] & Jeffrey A. Medin [2,9,23✉]

Enzyme and chaperone therapies are used to treat Fabry disease. Such treatments are expensive and require intrusive biweekly infusions; they are also not particularly efficacious. In this pilot, single-arm study (NCT02800070), five adult males with Type 1 (classical) phenotype Fabry disease were infused with autologous lentivirus-transduced, CD34+-selected, hematopoietic stem/progenitor cells engineered to express alpha-galactosidase A (α-gal A). Safety and toxicity are the primary endpoints. The non-myeloablative preparative regimen consisted of intravenous melphalan. No serious adverse events (AEs) are attributable to the investigational product. All patients produced α-gal A to near normal levels within one week. Vector is detected in peripheral blood and bone marrow cells, plasma and leukocytes demonstrate α-gal A activity within or above the reference range, and reductions in plasma and urine globotriaosylceramide (Gb$_3$) and globotriaosylsphingosine (lyso-Gb$_3$) are seen. While the study and evaluations are still ongoing, the first patient is nearly three years post-infusion. Three patients have elected to discontinue enzyme therapy.

[1] Department of Medical Genetics, Metabolics and Pediatrics, Alberta Children's Hospital, Cumming School of Medicine, Research Institute, University of Calgary, Calgary, AB, Canada. [2] University Health Network, Toronto, ON, Canada. [3] Department of Laboratory Medicine and Pathobiology, University of Toronto, Toronto, ON, Canada. [4] Department of Pathology and Laboratory Medicine, Western University, London, ON, Canada. [5] Department of Pediatrics, Western University, London, ON, Canada. [6] Children's Health Research Institute, London, ON, Canada. [7] Division of Medical Genetics, Department of Pediatrics, CIUSSS de l'Estrie-CHUS Hospital Fleurimont, Université de Sherbrooke, Sherbrooke, QC, Canada. [8] Department of Pathology and Molecular Medicine, McMaster University and Juravinski Hospital and Cancer Centre, Hamilton, ON, Canada. [9] Department of Pediatrics, Medical College of Wisconsin, Milwaukee, WI, USA. [10] Department of Oncology, McMaster University and Juravinski Hospital and Cancer Centre, Hamilton, ON, Canada. [11] Cancer Clinical Research Unit, Princess Margaret Cancer Centre, Toronto, ON, Canada. [12] Nova Scotia Health Authority, QEII Health Sciences Centre, Canadian Fabry Disease Initiative, Nova Scotia Fabry Disease Program, Halifax, NS, Canada. [13] Alberta Children's Hospital and Foothills Medical Centre, Calgary, AB, Canada. [14] Tom Baker Cancer Centre, Alberta Health Services, Calgary, AB, Canada. [15] Departments of Oncology and Medicine, Alberta Blood and Marrow Transplant Program, University of Calgary, Calgary, AB, Canada. [16] Department of Pathology and Laboratory Medicine, Cumming School of Medicine, University of Calgary, Calgary, AB, Canada. [17] Fred A. Litwin Family Centre in Genetic Medicine, Department of Medicine, University Health Network, Toronto, ON, Canada. [18] Division of Hematology, Department of Medicine, Dalhousie University, Halifax, NS, Canada. [19] Cumming School of Medicine, University of Calgary, Calgary, AB, Canada. [20] Rapa Therapeutics, Rockville, MD, USA. [21] University of Toronto, Princess Margaret Cancer Centre, Toronto, ON, Canada. [22] Division of Nephrology, Department of Medicine, Dalhousie University, Halifax, NS, Canada. [23] Department of Biochemistry, Medical College of Wisconsin, Milwaukee, WI, USA. [24]Deceased: Stephen Couban. ✉email: jmedin@mcw.edu

In Fabry disease, mutations of the X-linked *GLA* gene lead to accumulation of glycosphingolipids including globotriaosylceramide (Gb$_3$)[1,2] and globotriaosylsphingosine (lyso-Gb$_3$)[3,4]. This results in end-organ damage to the kidneys, heart, and brain leading to a decreased life expectancy[5,6]. Current approved treatments for Fabry disease include enzyme therapy (ET) and an oral pharmacologic chaperone (migalastat)[7–9]. Biweekly ET can reduce Gb$_3$ levels in urine, plasma, and tissues but is intrusive, not curative, and progressive disease continues to cause clinical symptoms and a decreased lifespan[10,11]. Moreover, antibody formation directed against the recombinant enzyme occurs, which may affect therapy outcome[10]. The short plasma half-life[12] requires biweekly infusions at considerable cost. Despite these issues, ET is recommended for treatment of Fabry patients to prevent progression in conjunction with nonspecific adjunctive therapies[13–15]. Migalastat is protein-variant specific, and therefore only available to a subset of Fabry patients with amenable mutations[8].

Gene therapy, in theory, would enable Fabry patients to receive a single treatment that could be more effective than current options and free them from ET. Transduced cell populations that continuously produce α-gal A may be more effective clinically. As well, cross-correction may make Fabry disease particularly amenable to gene therapy; enabling systemic correction with a lower number of vector transduced cells[16,17]. Transgenic mice, with tissue α-gal A activity >10,000 times endogenous levels, were healthy and did not have altered cyto-architecture;[18,19] thus high levels of α-gal A may not be deleterious.

In a pilot safety study, we have targeted enriched CD34+ hematopoietic stem/progenitor cells (HSPCs) for lentivirus (LV)-mediated gene therapy in patients with Fabry disease (NCT02800070, Fig. 1). Transduced HSPCs may deliver the functional enzyme to sites not accessible to ET. HSPC progeny may be a major source of Gb$_3$ and lyso-Gb$_3$; this may correct substrate accumulation at the point of origin. Use of autologous primitive HSPCs with conditioning may facilitate immune tolerance. We employed a recombinant LV with a self-inactivating LTR design and an optimized Kozak start sequence to deliver a human codon-optimized α-gal A transgene[20]. A single ex vivo LV transduction allowed for controlled dosing of the vector with a relatively low multiplicity of infection that should minimize insertional mutagenesis events[20]. We also utilized reduced-intensity melphalan conditioning, enabling outpatient management, fewer Grade 3 or 4 adverse events, and reduced cost. We report safety and outcome measures of the first gene therapy trial for Fabry disease.

## Results

**Patient enrollment**. Male patients with known Fabry disease from the Canadian Fabry Disease Initiative study of ET were approached for possible enrollment. Seven male Fabry disease patients previously treated with ET, ages 29–48 years, were enrolled (Table 1, Supplementary Table 1); two patients failed screening tests associated with inclusion and exclusion trial criteria. Patients were followed from January 2017 to February 2020 for this study. Ongoing follow-up will extend until February 2024.

**Study objectives**. The primary objectives of this study were to determine the safety and toxicity of autologous stem cell transplantation with mobilized CD34+ hematopoietic cells transduced with a lentiviral vector containing human codon-optimized α-gal A cDNA in adult male Fabry disease patients. Several secondary objectives were analyzed including monitoring levels of α-gal A activity in plasma, peripheral blood leukocytes and bone-marrow-derived mononuclear cells, and measuring levels of Gb$_3$ and lyso-Gb$_3$ in plasma and urine. In addition, the transduction efficiency of CD34+ hematopoietic cells was assessed as well as the presence and persistence of marked cells expressing α-gal A in peripheral blood. An exploratory objective was also incorporated to track the levels of anti-α-gal A antibodies in recipients. The primary endpoint of this study was toxicity as assessed using the

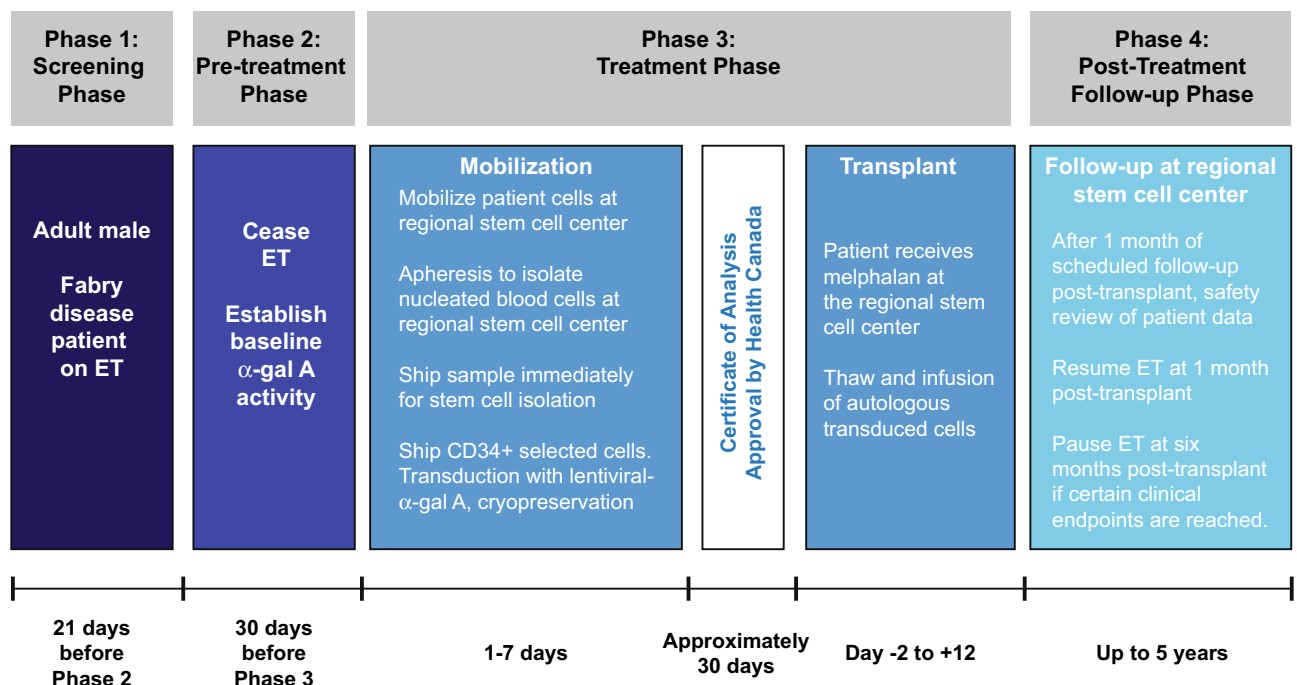

**Fig. 1 Study schema.** Five men with Type I Fabry disease were infused with autologous transduced CD34+-selected hematopoietic stem/progenitor cells engineered to express α-galactosidase A (α-gal A) following mild ablation. ET enzyme therapy.

**Table 1 Baseline patient demographics and treatment phase outcomes.**

| Parameter | Patient 1 | Patient 2 | Patient 3 | Patient 4 | Patient 5 |
|---|---|---|---|---|---|
| Age (years) | 48 | 39 | 39 | 37 | 29 |
| *GLA* mutation | p.Gln321Arg | p.Ser345Pro | p.Ala143Pro | p.Ala143Pro | p.Tyr134Ser |
| Age at diagnosis (years) | 36 | 29 | 0 | 4 | 14 |
| Age started ET (years) | 36 | 33 | 36 | 26 | 29 |
| *Fabry symptoms at baseline* | | | | | |
| Acroparesthesia | | | x | | |
| Angiokeratoma | x | x | | x | x |
| Cardiomyopathy | x | | x | x | |
| Chronic kidney disease | | x | | | |
| Cold intolerance | | x | | | |
| Corneal verticillata | | x | x | x | |
| Gastrointestinal | x | x | | x | x |
| Heat intolerance | | x | | | |
| Hypertension | | | | x | x |
| Hypohidrosis | | x | | x | x |
| Headaches and migraines | | | | | x |
| Pain | | | | x | x |
| Peripheral sensory neuropathy | | x | | | |
| Proteinuria | x | x | | | |
| Tinnitus or hearing loss | | | x | x | x |
| Mobilization agents | G-CSF | G-CSF + plerixafor | G-CSF + plerixafor | G-CSF + plerixafor | G-CSF |
| Apheresis yield CD34$^+$ cells ×10$^6$/kg | 8.4 | 9.2 | 18.1 | 9.0 | 5.1 |
| Drug product VCN (copies/genome) | 0.68 | 1.43 | 0.81 | 1.37 | 1.13 |
| Drug product infused: CD34$^+$ cells ×10$^6$/kg | 4.9 | 6.4 | 13.8 | 6.2 | 3.1 |
| Colony PCR[a] | | | | | |
| Mock | 0% | 0% | 0% | ND | ND |
| LV-AGA (Drug Product) | 52% | 62% | 54% | ND | ND |
| Colony PCR on Day −2 BM samples | ND | 0/86 (0%) | 0/88 (0%) | 0/92 (0%) | 0/88 (0%) |
| Colony PCR on Day 28 BM samples | 10/28 (35.7%) | 70/92 (76.1%) | 63/90 (70.0%) | 52/92 (56.5%) | 56/88 (63.6%) |
| Actual study day for colony PCR | Day 33 | Day 28 | Day 28 | Day 28 | Day 32 |
| Day −2 BM α-gal A nmoles/hr/mg | 0.3 | 1.0 | 1.6 | 1.0 | 0.8 |
| Day 28 BM α-gal A nmoles/hr/mg | 220 | 350 | 169 | 320 | 569 |
| Time to engraftment (absolute neutrophil count >0.5 x 10$^9$ cells per L)[b] | Day 12 | Day 13 | Day 11 | Day 11 | Day 12 |
| Time to engraftment (platelet count >20 x 10$^9$ per L) | Day 12 | Day 12 | Day 12 | Day 11 | Day 12 |
| Day −2 BM VCN copies/genome | 0.00 | 0.00 | 0.00 | 0.00 | 0.00 |
| Day 28 BM VCN[c] copies/genome | 0.47 | 0.89 | 0.33 | 1.10 | 1.21 |
| Maximum PB VCN copies/genome | 0.55 | 1.10 | 0.78 | 0.65 | 1.30 |
| First Day Plasma α-Gal A activity observed | Day 8[d] | Day 7 | Day 6 | Day 6 | Day 7 |
| Days from Cell Infusion to ET withdrawal | +548 | ND | −41[e] | +214 | ND |

*BM* bone marrow, *ET* enzyme therapy, *ND* not done, *PB* peripheral blood, *PCR* polymerase chain reaction, *VCN* vector copy number.
[a]Transduction efficiency—colony PCR assays.
[b]Ref. 35.
[c]Day 28–33.
[d]Defined by activity >2 times the average of pretransplant values.
[e]Patient 3 discontinued ET prior to infusion and chose not to resume.

National Cancer Institute of Canada (NCIC) Common Terminology Criteria for Adverse Events (CTCAE), Version 4.03. Because of the small sample size of this clinical study, there was insufficient power for comprehensive statistical analysis.

**Mobilization and leukapheresis.** During the pretreatment phase, ET was stopped for a minimum of 30 days prior to transplant. ET was stopped so that we could obtain a consistent cohort of baseline measurements. After cessation of ET, peripheral blood (PB) CD34$^+$ HSPCs were mobilized using filgrastim in Patients 1 and 5, and filgrastim and plerixafor in Patients 2–4. Leukapheresis yielded 5.1–18.1 x 10$^6$ CD34$^+$ cells/kg, with a final total number of 3.65–8.35 x 10$^8$ CD34$^+$ cells (Table 1). The drug product yield was 3.1–13.8 x 10$^6$ CD34$^+$ cells/kg (Table 1).

**Treatment phase.** Enzyme activity, colony PCR, and VCN assays were performed on the drug product (Table 1). Prior to infusion,

patients were administered a single dose of melphalan IV at 100 mg/m$^2$ on Day −1. On Day 0, autologous CD34+ transduced cells were infused. Filgrastim (5 µg/kg) was administered subcutaneously daily from Day 5 until neutrophil count reached ≥1.5 x 10$^9$ cells/L. All patients (except Patient 3) restarted ET 30 days after infusion. According to a Health Canada-approved protocol amendment, all patients are eligible to discontinue ET. Two patients subsequently discontinued ET (Patient 1 at day 548, Patient 4 at day 214 after transplantation, respectively). A third patient (Patient 3) chose not to resume ET after transplantation.

**α-gal A enzyme activity.** Circulating α-gal A activity was first detected in all patients between Days 6 and 8 following infusion and attained reference range levels in all patients (Fig. 2a). Leukocyte α-gal A reached specific activity levels above the reference range (Fig. 2b). Both plasma α-gal A

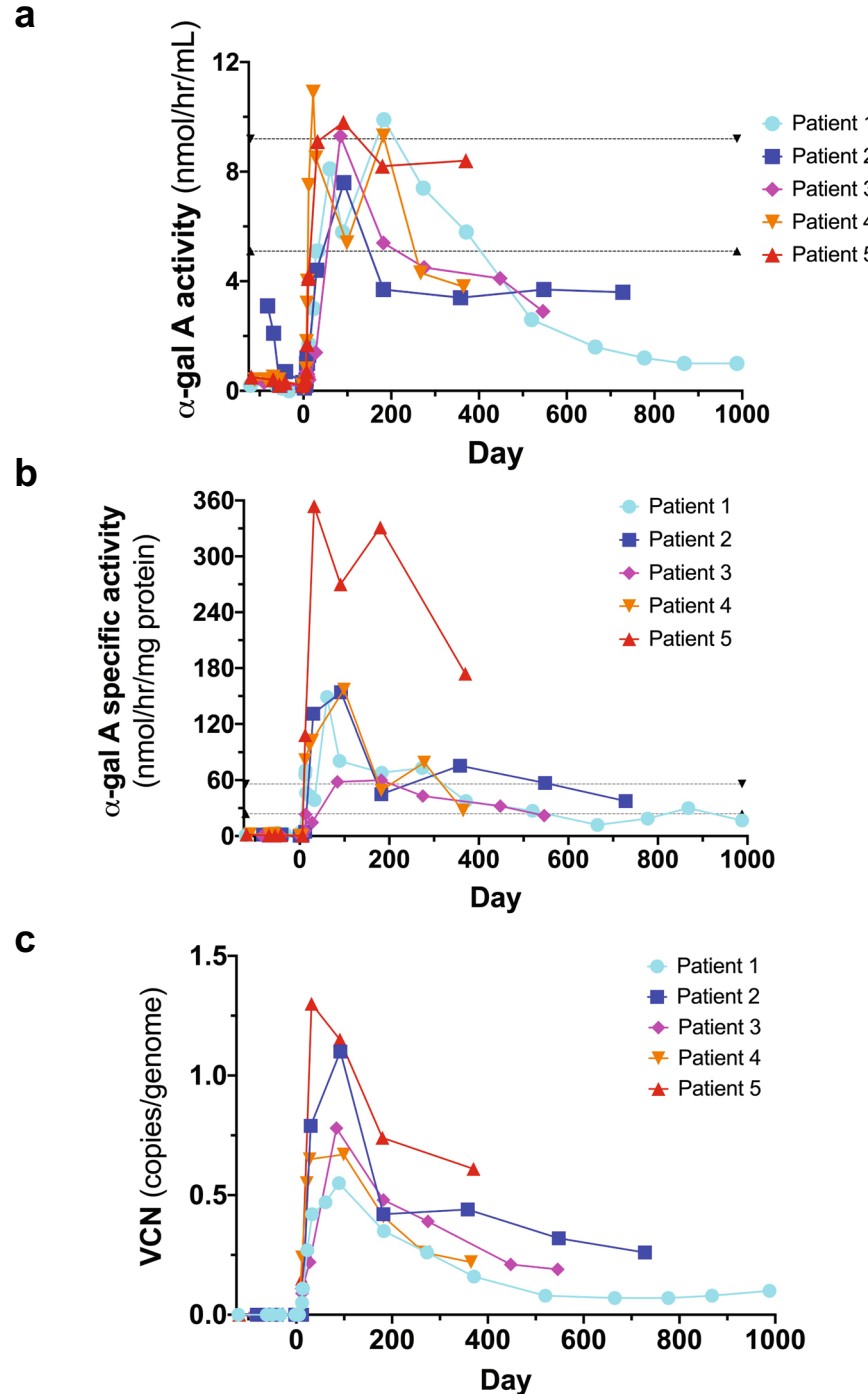

**Fig. 2 α-galactosidase A (α-gal A) enzyme activity and vector copy number (VCN). a** Plasma α-gal A activity attained reference range levels in all patients; although decreased over time, the plasma α-gal A enzyme activity levels are above what is observed in Fabry disease patients and have not returned to original baseline levels. The reference ranges (dotted lines) were defined by Dr. Rupar's laboratory based on 150 specimens referred for diagnostic testing. Males with classic Fabry disease have plasma levels around 1 nmol/h/ml. **b** Leukocyte α-gal A attained supranormal specific activity levels for each patient. Although decreased over time, leukocyte α-gal A-specific enzyme activity levels are above what is seen in Fabry disease patients and have not returned to original baseline levels. The reference ranges (dotted lines) were defined by Dr. Rupar's laboratory based on 150 specimens referred for diagnostic testing. **c** VCN in peripheral blood reached between 0.55 and 1.10 copies/genome in all patients, and although decreased over time, has remained above 0.05 copies/genome in all patients to date (almost 3 years in Patient 1).

enzyme activity levels and leukocyte α-gal A specific enzyme activity levels decreased over time; yet the activity levels are above Fabry disease patients[21] and have not returned to original baseline levels (Fig. 2a, b)[22]. Plasma and leukocyte α-gal A-specific activities mirrored each other (Supplementary Fig. 1).

**Vector copy number.** The infused drug product vector copy number (VCN) ranged from 0.68 to 1.43 (copies/genome) (Table 1). The VCN ranged from 0.33 to 1.21 (copies/genome) in bone marrow (BM) aspirates obtained from each patient at one month (Table 1). PB VCN reached a range of 0.55 to 1.10 copies/genome in all patients and has decreased over time but has

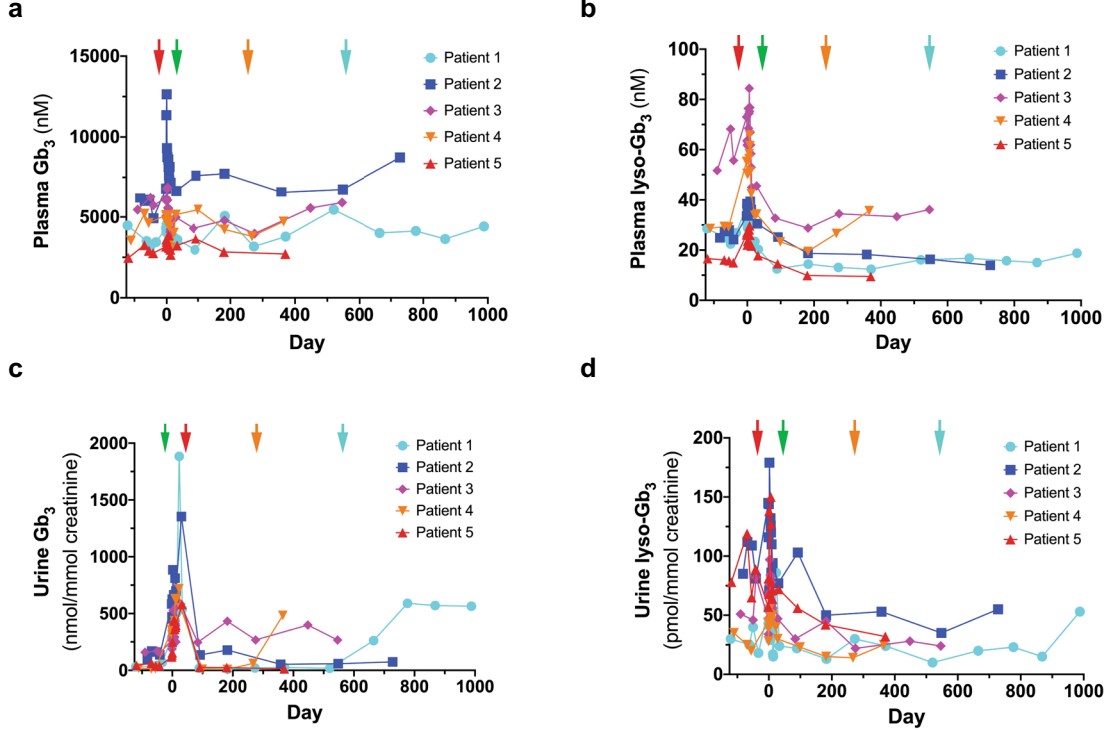

**Fig. 3 Total plasma and urine globotriaosylceramide (Gb3) and lyso-Gb3 levels.** Plasma Gb3 (**a**), plasma lyso-Gb3 (**b**), urine Gb3 (**c**), and urine lyso-Gb3 (**d**) levels are illustrated for each patient. Red arrow is at Day −30 when ET was stopped prior to mobilization. Green arrow is at Day 30 when ET was restarted for Patients 1, 2, 4, and 5. The orange arrow demarks when Patient 4 stopped ET at Day 214 and the blue arrow is when Patient 1 stopped ET at Day 548. Patient 3 chose not to restart ET.

remained above 0.05 copies/genome in all patients (almost 3 years in Patient 1) (Table 1, Fig. 2c). VCN mirrored leukocyte α-gal A-specific activity (Supplementary Fig. 2).

**Plasma and urine Gb$_3$/lyso-Gb$_3$ levels**. Total plasma and urine Gb$_3$ levels were variable, especially during the Treatment Phase. Plasma and urine lyso-Gb$_3$ levels were reduced over time in most patients (Fig. 3). This reduction was sustained after discontinuation of ET in Patient 3, with the exception of urine Gb$_3$. Increases in urine Gb$_3$ levels were also observed when Patients 1 and 4 stopped ET treatment. Urine lyso-Gb$_3$ also increased after Patient 1 stopped ET treatment. Plasma Gb$_3$ levels in Patient 3 fluctuated after infusion, but rose at later time points. Plasma Gb$_3$ and plasma lyso-Gb$_3$ have increased in Patient 4 after this patient chose to stop ET. Data for the Treatment Phase for all parameters shown in Figs. 2 and 3 are illustrated in Supplementary Fig. 3.

**Clinical parameters**. Patients initially experienced a drop in weight during the Treatment Phase. In Patients 1–3 and 5, weights increased over time following transplantation (Fig. 4a). eGFR increased during the Treatment Phase in all patients (Fig. 4b). Following gene therapy, eGFR returned to baseline levels and was relatively stable in all patients, except for Patient 2. This patient displayed progressive chronic kidney disease with significant proteinuria (Table 1) during screening. Linear regression demonstrated that Patient 2 had the steepest eGFR slope, whereas the eGFR slope was near zero for the remaining patients (Supplementary Fig. 4). Proteinuria was also described in Patient 1 during screening (Table 1); 24-h urinary protein measurements confirmed this observation in Patients 1 and 2 (Fig. 4c). Patient 1 was described with left ventricular hypertrophy during screening (Table 1). Monitoring of troponin levels suggests that Patient 1 is displaying cardiac features of Fabry

disease (Fig. 4d). Left ventricular mass index (LVMI) monitoring by MRI and ECHO revealed that cardiac hypertrophy observed in Patient 1 was relatively stable for nearly three years post-infusion (Fig. 4e). LVMI was also stable for Patients 2, 4, and 5 during the study period.

**Anti-α-gal A antibody titer**. Anti-α-gal A antibody (IgG) levels increased for Patient 1 within 6 months post-infusion, before gradually declining (Fig. 5). Patient 3 had the highest titer at screening, which was maintained through mobilization and the Treatment Phase but decreased after 6 months. Patients 4 and 5 had low titers at screening, which dropped by mobilization (Patient 5) or after the Treatment Phase (Patient 4). Patient 2 had no detectable anti-α-gal A antibody.

**Safety monitoring**. No unexpected trends or safety events have been identified. The safety profile is consistent for patients undergoing melphalan conditioning for autologous hematopoietic cell transplantation (Supplementary Table 2). Two AEs (nausea [Grade 1] and cough [Grade 2]) were possibly related to the investigational product. There were 20 AEs reported of Grade 3 or 4 (Supplementary Table 2). Patient 2 experienced anorexia over 51 days corresponding to a lack of appetite (Supplementary Table 2) and this was assessed to be related to study procedures. All AEs Grade 3 or higher were unrelated to protocol treatment, but were related to the study procedures. Two SAEs were related to the study procedures (febrile neutropenia in Patient 4 and peripherally inserted central catheter line infection/thrombosis of the right arm in Patient 5).

## Discussion

In this pilot clinical trial of LV-mediated gene therapy in 5 men with Type 1 (classical) Fabry disease, all patients demonstrated a

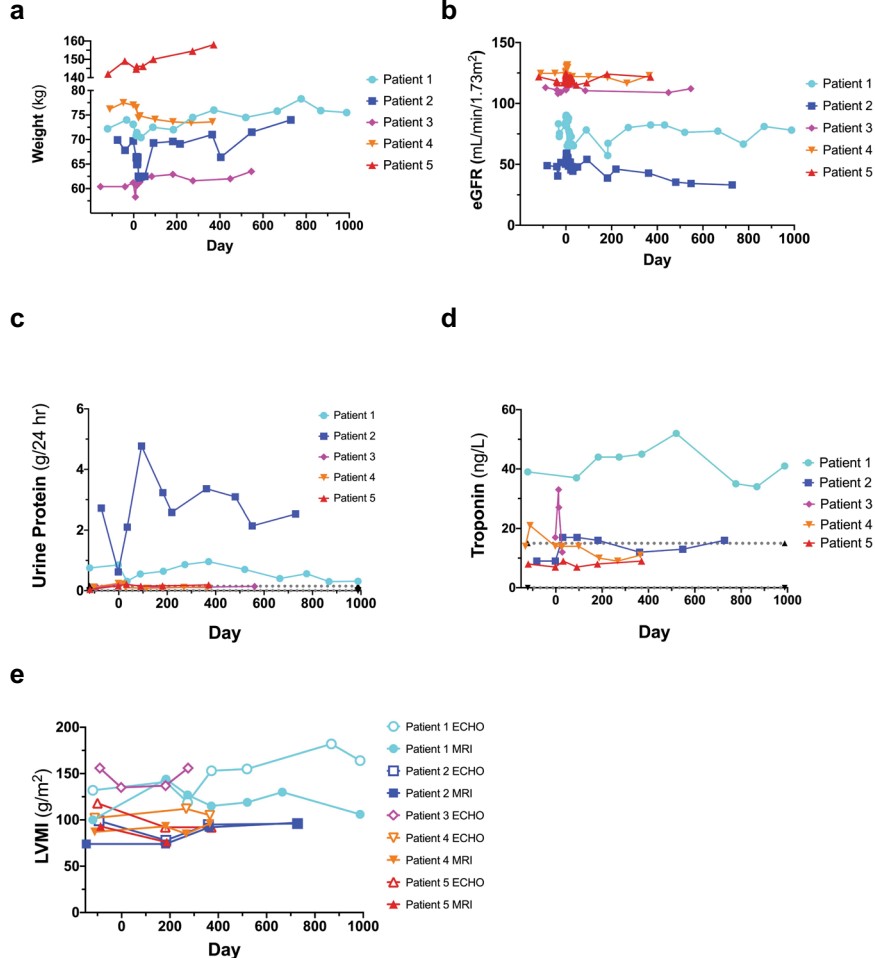

**Fig. 4 Clinical parameters.** Weight (**a**), estimated Glomerular Filtration Rate (eGFR) (**b**), urinary protein secretion (**c**), troponin (**d**), and Left Ventricular Mass Index (LVMI) (**e**) was monitored during the course of the trial for all patients. eGFR was calculated using the Chronic Kidney Disease-Epidemiology Collaboration formula. Values from Magnetic Resonance Imaging (MRI) and Echocardiography (ECHO) are shown for LVMI (**e**). Patient 3 did not attend all cardiac assessments and has been omitted from Fig. 4e.

sustained safety profile. One patient is now out nearly 3 years from his infusion date. We chose a reduced-intensity conditioning regimen for drug product engraftment, based on many years of transplant experience with melphalan in Toronto. Since there is limited experience with transplant conditioning procedures in Fabry patients, and Fabry disease patients have considerable comorbidities, we also wanted to establish whether mobilization and engraftment could be tolerated in this population. All patients received a low melphalan dose; 3 men (Patients 1, 4, and 5) received the treatment as an outpatient and returned home the same day. Full ablation regimens may require lengthy hospital stays and result in additional AEs[23–27].

We demonstrated efficient LV-mediated gene transfer into enriched Fabry patient CD34+ cells. VCN levels were modest, minimizing the chance of genotoxicity. Our regimen led to increased circulating and intracellular α-gal A activity. All patients reached reference range levels. Values declined over time, possibly due to an exhaustion of transduced short-term repopulating HSPCs, but remained above what is typically observed in Fabry patients and well above pretreatment levels in all five cases. Whether the enzyme values reach an asymptote reflecting engraftment of transduced long-term HSPCs remains to be determined. VCN and α-gal A activities were well correlated, underscoring the fidelity of our therapeutic LV construct. Patient weight, eGFR, proteinuria and LVMI were stable for all patients

throughout the study period (ranging from 12-33 months post-infusion), with the exception of Patient 2 who is showing symptoms of chronic kidney disease associated with his Fabry disease. Plasma lyso-Gb$_3$ levels decreased in all patients with the exception of Patient 4 where an increase was observed. Plasma Gb$_3$ levels were generally stable in all patients, except for increases at later time points in Patients 2 and 3. Urine lyso-Gb$_3$ values were increased in Patient 1, stable in Patient 4, and decreased in Patients 2, 3 and 5. Finally, urine Gb$_3$ levels were stable in Patients 2 and 5 who continued ET after gene therapy, but were observed to increase in Patients 1, 3, and 4 who elected to withdraw from their ET infusions. Interestingly, Patient 3 who chose to stop ET prior to mobilization had sustained low levels of plasma lyso-Gb$_3$ and urine lyso-Gb$_3$; a result of the gene therapy product alone.

It is possible that a more myelo-suppressive conditioning regimen would result in better engraftment of LV/AGA transduced cells—although this hypothesis remains to be tested. In selection of a partial myeloablative conditioning regimen, we considered that our prospective patients were relatively healthy males with Fabry disease who were receiving ET prior to recruitment to our trial. Unlike most other acute disorders that have been treated by gene therapy, Fabry disease is a chronic disease. As such, our clinical team and our regulatory body were reluctant to implement a full myeloablative conditioning regimen.

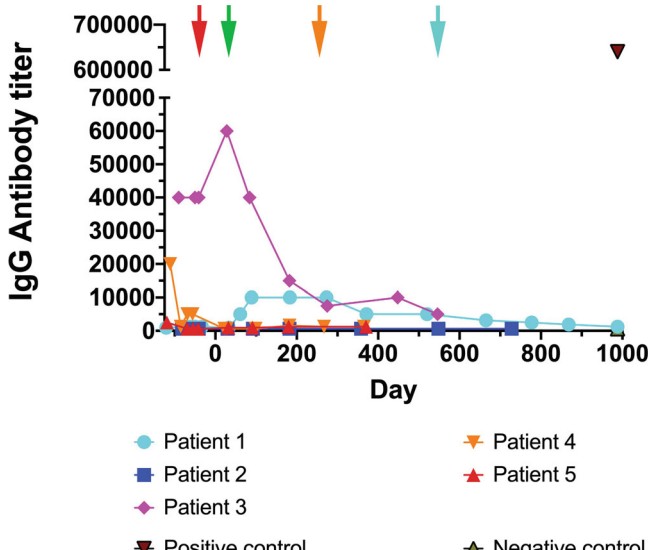

**Fig. 5 Immunoglobulin G (IgG) antibody titer.** Enzyme-Linked Immunosorbent Assays (ELISA) were completed at each time point as shown. In four patients who had detectable anti- α-galactosidase A (α-gal A) antibodies, levels declined in three patients and rose in one patient, although the titers never reached the levels of a positive control sample collected following enzyme therapy (ET). Red arrow is at Day −30 when ET was stopped prior to mobilization. Green arrow is at Day 30 when ET was restarted for Patients 1, 2, 4, and 5. The orange arrow demarks when Patient 4 stopped ET at Day 214 and the blue arrow is when Patient 1 stopped ET at Day 548. Patient 3 chose not to restart ET.

Such protocols are more invasive, are accompanied by a greater number of adverse events, and have the potential for future oncogenic events. Likewise, infection of our target cells at a higher MOI or multiple infection courses may also lead to higher transgene expression; however, GMP-grade, recombinant LV was already limiting in our trial.

Host immunity to ET is thought to limit therapeutic efficacy; and, it is unknown whether gene therapy might exacerbate anti-α-gal A antibody responses. Each of the three patients (Patients 3–5) who had pre-existing anti-enzyme IgG antibodies had nearly complete elimination of titers after the gene therapy intervention without resurgence of levels despite continuous enzyme exposure from the transduced cell-derived source. Only one patient (Patient 1) had an onset of antibody titer to enzyme, which diminished over time. Two patients (Patients 2 and 5) remained antibody titer negative after gene therapy. We conclude that gene therapy is not only not highly immunogenic, but may also reduce pre-existing immunity to foreign ET.

If successful, LV-mediated gene therapy may result in beneficial outcomes for Fabry patients with a single treatment. No serious safety concerns were observed in our pilot trial. All Fabry patients in this study were responsive to the LV-mediated gene therapy at some level. Three patients have discontinued ET to date. Recipients had detectable levels of VCN in PB and BM, plasma and leukocyte α-gal A-specific activity above or within the reference range, and reductions in Gb$_3$ and lyso-Gb$_3$ levels. Gene therapy may be an effective treatment option in patients with Fabry disease but requires more study.

## Methods

**Patients.** This multiple-center, single-arm, Phase 1 trial recruited patients from Calgary, AB; Halifax, NS; and Toronto, ON; from September 2016–October 2018. Prior to recruitment to this study, volunteers were screened and provided written informed consent. All patients consented to the release of deidentified data in this report. Patients were assigned a unique study identifier by Ozmosis Research Inc.

All data presented within this manuscript have deidentified each patient and regional stem cell center. Recruitment (from September 2016 to October 2018) was restricted to men (age 18–50) with confirmed Type 1 (classical) phenotype Fabry disease (with *GLA* genotyping) who had received ET for at least 6 months prior to study enrollment. Inclusion criteria comprised an estimated glomerular filtration rate (eGFR), > 45 mL/min/1.73 m$^2$ (chronic kidney disease-epidemiology collaboration equation [CKD-EPI]) and left ventricular ejection fraction >45%. Patients with advanced Fabry disease were excluded. Additional eligibility criteria can be found in Supplementary Table 3. Criteria for ET withdrawal is indicated in Supplementary Table 4. The current version of the Health Canada-approved clinical protocol can be accessed in the Supplementary Data.

**Lentiviral vector.** The LV-AGA vector has been described previously[20]. Large-scale, high-titer clinical-grade, recombinant lentivector was manufactured, purified, and qualified by the Indiana University Vector Production Facility, Indianapolis, IN. A total of five patients were treated in this Phase I safety trial due to the quantity of the lentiviral vector produced for this study.

**Study design.** Patients were recruited to three regional hematopoietic stem cell centers (Calgary, AB; Toronto, ON; and Halifax, NS). All steps of mobilization, apheresis, conditioning, and transplantation occurred at the relevant site for each patient. Patients were initially mobilized with filgrastim (granulocyte-colony stimulating factor [G-CSF] 16 µg/kg). If peripheral blood CD34$^+$ counts were low (i.e., if the predicted CD34$^+$ count was <50% of our target of $12.5 \times 10^6$/kg), filgrastim was supplemented with plerixafor (240 µg/kg). Three patients were mobilized with filgrastim and plerixafor; for one patient this occurred because of a site preference for this regimen. A backup graft of at least $2.5 \times 10^6$ unmanipulated CD34$^+$ cells/kg was harvested from each patient for utilization in the event of graft failure. After apheresis, the nucleated cells were transported to the Juravinski Cancer Centre, Hamilton, ON, for CD34$^+$ cell enrichment. Cells were then transported to the Orsino Cell Processing Laboratory, Princess Margaret Cancer Centre, Toronto, ON, for the lentiviral transductions. The transduction protocol was described in detail[20]. Quality control was performed: cell viability, evaluation of α-gal A activity in the transduced CD34$^+$ cell population, and VCN were evaluated. After cryopreservation and safety/sterility testing, and an approved Certificate of Analysis from Health Canada, the drug product was transported back to the stem cell center of origin. Patients received melphalan (100 mg/m$^2$) intravenously one day prior to autologous drug product infusion (Fig. 1). After we observed prolonged α-gal A activity in both leukocytes and plasma of Patient 1, the protocol was amended to allow patients to discontinue ET after an appropriate consultation/consent procedure. All patients are eligible to discontinue ET, and three patients have elected to stop ET at time of manuscript submission.

**Safety and functional efficacy assessments.** Safety endpoints included reporting of adverse events (AEs) and serious adverse events (SAEs), scored by the National Cancer Institute Common Terminology Criteria for Adverse Events version 4.03. Analysis of drug product efficacy included the detection of plasma, leukocyte, and BM mononuclear cell (BMMC) α-gal A activity[20,28,29]; analysis of Gb$_3$ and lyso-Gb$_3$ from plasma and urine by tandem mass spectrometry (MS/MS)[30–32]; presence and persistence of LV marked cells in PB[20]; transduced drug product and BM HSPC as measured by colony polymerase chain reaction (PCR) assay[20]; anti-α-gal IgG antibody levels measured by enzyme-linked immunosorbent assay (ELISA) (modified from Lee)[33]; clinical outcomes; and detection of VCN. The primers utilized for quantitative PCR are illustrated in Supplementary Table 5. All PB enzyme activities were measured in the trough period more than 10 days after ET administration and during the 60 days bracketing the infusion of transduced cells when ET was stopped. The reference ranges for plasma α-gal A activity and leukocyte α-gal A specific activity were defined by London Health Sciences Centre Clinical Biochemical Genetics Laboratory based on 150 specimens referred for diagnostic testing. Cardiac echocardiograms and magnetic resonance imaging was performed as described[34].

**Study oversight.** The trial (NCT02800070, Health Canada-approved April 26, 2016) was conducted in compliance with the Declaration of Helsinki and local institutional and/or university Human Experimentation Committee requirements. Research Ethics Board (REB) approval was provided by University Health Network Research Ethics Board, Alberta Health Services Research Ethics Board, Capital Health Services Research Ethics Board, Hamilton Integrated Research Ethics and the Medical College of Wisconsin Institutional Review Board. This study was designed by the Fabry Disease Clinical Research and Therapeutics (FACTs) team. Clinical trial management was performed by Ozmosis Research, Inc., of Toronto, Canada. Clinical data were collected by the investigators of the FACTs team who were responsible for clinical assessments during the Treatment Phase and long-term follow-up. Clinical and laboratory data were reviewed monthly by the FACTs team. Clinical decisions and safety monitoring for each patient were completed by a Clinical Trial Steering Committee (CTSC) of the FACTs team. Safety review meetings were conducted by the CTSC 1 month after the transplant date for each patient. The next patient was then eligible for recruitment to the study only after the CTSC reviewed all of the safety data and unanimously concluded that there

have been no treatment-related serious adverse events. A Data Monitoring Safety Committee (DSMC) reviewed clinical and safety data after the Treatment Phase for Patients 1 and 3. Interim analysis and publication of this clinical trial was authorised by the DSMC. As this is the first lentivirus-directed gene therapy trial to focus on adults with a lysosomal storage disorder, the DSMC supported reporting of safety data up to 1-year post-infusion of each patient.

**Reporting summary**. Further information on research design is available in the Nature Research Reporting Summary linked to this article.

## Data availability

The datasets generated during and/or analyzed during the current study are available from the corresponding author on reasonable request. Source data are provided with this paper.

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

## Acknowledgements

We thank the patients for their commitment to the project. We thank Cindy Yau, Rupi Mangat, Sarah Young, and Pam Degendorfer from Ozmosis Research, Inc., for their support of the trial. From the Cumming School of Medicine, University of Calgary, we acknowledge Andrew Daly and John Klassen for inpatient care to all stem cell transplant recipients. We acknowledge Kathleen Estes who is managed by Donna Simcoe of Simcoe Consultants, Inc. for help with medical writing. Funding for writing support was provided by AVROBIO, Inc who did not contribute to the content and only provided an administrative review of the final manuscript. We made final decisions regarding all content. The following institutions provided funding for this study: Canadian Institutes of Health Research (CIHR, grant number 119187), The Kidney Foundation of Canada, and the MACC Fund. Financial support for the study was also provided by AVROBIO, Inc.

## Author contributions

α-gal A enzymatic assays were performed by the London Health Sciences Centre Clinical Biochemical Genetics Laboratory, London, ON (J.W.R. and C.A.R.). Gb₃ and lyso-Gb₃ analyses were performed at the Université de Sherbrooke, Sherbrooke, QC (M.B. and C.A.-B.). Vector copy number assays and ELISAs to measure anti-α-gal IgG antibodies were performed at the University Health Network, Toronto, ON (J.H., K.G., D.L.B., and J.A.M.), and the Medical College of Wisconsin, Milwaukee, WI (K.G., W.M.M., and J.A.M.). The FACTs team had confidential access to the data through the trial management company. The sponsor (University Health Network) played no role in the study design, data collection and analyses, or drafting of the manuscript. A.K., D.L.B., J.H., C.A.R., C.A.-B., W.M.M., D.H.F., A.K., M.L.W., R.F., and J.A.M. participated in research design. Research was performed by A.K., D.L.B., J.H., C.A.R., J.W.R., C.A.-B., M.B., P.O., K.G., W.M.M., G.F., S.W., K.L., S.J., A.C., N.P., C.F.M., S.C., P.R.D., D.H.F., A.K., M.L.W., R.F., and J.A.M. Data were analyzed by A.K., D.L.B., J.H., C.A.R., J.W.R., C.A.-B., M.B., K.G., W.M.M., D.H.F., A.K., M.L.W., R.F., and J.A.M. J.A.M. and D.L.B. wrote the manuscript and it was reviewed by A.K., C.A.R., C.A.-B., D.H.F., A.K., M.L.W., and R.F. All authors approved the final manuscript before publication.

## Competing interests

A.K. received grants, consulting fees, revenue distribution agreement, speaker fees and travel support with AVROBIO, Inc. as well as revenue distribution agreement with University Health Network regarding gene therapy using technology from this work. D.L.B. and J.H. were partially paid from a Sponsored Research Agreement—AVRO-BIO, Inc. C. A. Rupar has the following financial relationships to disclose: the Biochemical Genetics clinical diagnostic laboratory at his home institution is contracted by AVROBIO, Inc. to assay enzymes on a fee for service basis. He is the laboratory director but receives no personal compensation. C.A.-B. has received a service contract and honoraria for biomarker analysis with AVROBIO, Inc., grant from CIHR. K.G. had travel paid for by AVROBIO, Inc. S.W. has received nonfinancial support from Sanofi-Genzyme, nonfinancial support from Takeda Pharmaceuticals (formerly Shire HGT), personal fees and nonfinancial support from Amicus Therapeutics. K.L. has received travel grant and honorarium from Amicus Therapies; travel grant and speaker fees from Sanofi-Genzyme; travel grant, consulting fees and speaker fees from Takeda Pharmaceuticals; medical advisor to the Canadian Fabry Disease Association. C.F.M. has received grants, personal fees, and nonfinancial support from Takeda Pharmaceuticals (previously Shire HGT), grants, personal fees and nonfinancial support from Sanofi-Genzyme, nonfinancial support from Amicus Therapeutics. A.K. has received consultancy fees from AVROBIO, Inc. unrelated to this study. M.L.W. has received research grants, consulting fees, speaker fees and travel support with Amicus Therapeutics, Protalix, Sanofi-Genzyme and Takeda, revenue distribution agreement with University Health Network regarding gene therapy using technology from this work. J.A.M. has the following financial relationships to disclose: SAB—Rapa Therapeutics. Honoraria—Sanofi-Genzyme, Shire. Co-Founder—AVROBIO, Inc. Shareholder—AVROBIO, Inc. Grants from Canadian Institutes of Health Research and Kidney Foundation of Canada and AVROBIO, Inc. M.B., A.C., P.R.D., R.F., D.H. F., G.F., S.J., W.M.M., P.O., N.P., and J.W.R. have no financial relationships to disclose in relation to this trial.
