## [Peer Review File · Nature Communications]

Reviewers' Comments:

Reviewer #1:

Remarks to the Author:

The authors have clarified most of my concerns. Nevertheless, in opinion of this reviewer some information related to insertional site analyses would be necessary to verify important safety aspects of this first Phase I gene therapy trial in Fabry patients.

I still think that statements indicating that this gene therapy protocol results in sustained correction responses need to be tempered, since in several instances this is not evident.

Additionally I strongly recommend to include some discussion about possibilities for improving this approach of gene therapy in Fabry disease. For example a paragraph mentioning advantages and limitations of giving a more myelo-suppressive conditioning would be useful. Similarly, the authors mention that a low number of VCN/cell is an advantage, although this reviewer considers that it would be convenient to increase this VCN to facilitate a higher release of the enzyme, and thus to achieve a higher therapeutic efficacy.

Reviewer #2:

Remarks to the Author:

In this report, Kahn et al. present the results of a Phase 1, pilot study of lentiviral-mediated gene therapy for Fabry disease. The investigators enrolled and treated 5 subjects and were able to obtain successful transduction of CD34+ hematopoietic progenitors that were infused after reduced intensity conditioning with melphalan. The procedure resulted in the appearance of gene-corrected peripheral blood cells and the intracellular and plasma production of alpha-Gal A activity in all patients, which demonstrated the feasibility of the approach. These results are encouraging and support further development and application of the procedure.

Major comments

In the abstract and on pages 9-10, the sentences "all patients demonstrated a sustained corrective response" and "All patients had... a reduction in plasma and urine Gb3 levels, and plasma and urine lyso-Gb3 levels" appear to be over-interpretations of the results. It would seem that "corrective" effects should be only considered for patients 1, 3, and 4 who stopped ET. While it is true that treatment led to increased a-gal A activity in all 5 subjects, Pt. 1 showed an increase of urine Gb3 and lyso-Gb3 after ET stop, Pt. 3 showed levels of plasma and urine Gb3 at the last determination that appear to be higher than pre-gene therapy baseline values, and Pt. 4 showed an increase of plasma and urine Gb3 and lyso-Gb3 after ET stop. In addition, effects of the treatment on the clinical parameters shown in Figure 4 are not clearly evident. Based on these observations, it seems warranted that conclusions regarding any corrective effects that may have been observed be significantly tempered.

Minor issues

Page 5: the Authors should consider rephrasing the sentence "Gene therapy enables Fabry patients to receive ..." to indicate that, if successful, gene therapy may result in beneficial results with a single treatment.

Page 5: the sentence "A single ex-vivo LV infection..." should be changed to "A single ex-vivo LV transduction..."

Page 6: was ET stopped before treatment to provide selective advantage to gene-corrected cells?

Figure 2a: a tentative explanation for the progressive reduction of alpha-gal A activity and VCN

should be added to the Discussion, especially addressing the possibility of toxicity of high levels of alpha-gal A expression or high LV copy numbers.

Page 7: please clarify the meaning of "(i.e. prior to mobilization)".

Figure 5: the reference arrows are missing.

Methods, page 11: please specify what was considered a "low" blood CD34+ count.

Methods: a brief description of the transduction conditions (schedule, growth factors, MOI) would be of interest.

Reviewer #3:

Remarks to the Author:

Lentivirus-Mediated Gene Therapy for Fabry Disease

The authors have revised their original manuscript, which is improved in several ways. However, there are still some major issues.

Cross-correction

Critical for gene-therapy in Fabry disease using this approach is the possibility that cross corrections occurs, since the primary storage is not in the hematopoietic system. The authors point out that in the studies by Medin and Huang, cross correction was shown. However, the construct used by Medin et al resulted in an enzyme that could be taken up in a mannose-6-phosphate dependent way: the same as the recombinant enzymes used for ERT. However as elegantly discussed by Beck and Cox in a recent paper, (Mol Genet Metab Rep. 2019 Dec; 21: 100529.) the molecular human form of α -galactosidase in plasma is the mature 46 kDa enzyme and not the high-uptake, mannose 6-phosphorylated form. In the Huang paper, it is not clear how the enzyme is taken up and whether there is (sufficient) man-6-phosphorylation. Indeed, an increase in AGalA activity in PBMC's and plasma is shown (since the hematopoietic system is transduced), but hardly in tissues: only in organs that contain many hematopoietic cell line derived cells such as macrophages (spleen and liver) but not in heart and kidney, the crucial organs to target in Fabry disease. What is more, no significant Gb3 clearance was shown in these organs.

Biochemistry

Another major issue is the response in terms of biochemical parameters: plasma and urine globotriaosylceramide (Gb3) and globotriaosylsphingosine (lyso-Gb3). In the abstract, reductions in plasma and urine Gb3 and lysoGb3 are mentioned as the most robust outcome measures. However, the data show different and perhaps even non-sustainable results. Surprisingly, in their rebuttal they argue that there is no clinical correlate with these biochemical parameters, and their argument to use them is "to keep with current literature". This is not a valid argument. Either you are convinced that these parameters are of value, which I believe is true (in the sense that successful enzyme therapy in classical males is always followed by a decline in Gb3 and lysoGb3, which is reversed with dose interruptions or antibody formation), or you do not believe that this is a valid parameters, but then you should not use it as the most important clinical endpoint.

Clinical outcomes

Following criticism on the course of the biochemical parameters and lack of clinical data, the authors have added new data (weight, eGFR, urinary protein, troponin, and LVMI). Unfortunately, these data do not support the conclusion that gene therapy is effective. They state that weight is increasing (which is obviously a parameters that can be influenced by many factors), however, there is hardly any change in weight, with the exception of one obese patient adding more weight (nr 5). Most patients appear to have very limited organ damage. That proteinuria and eGFR are relatively stable in those with preserved kidney function is not exceptional. In patient 2, with renal impairment, there is a gradual further decrease in eGFR, consistent with observations during ERT. The other parameters are all over the place, due to variations in measurements, I assume, which is frequently observed in these patients. Hence, no support for clinical benefit of gene therapy can

be concluded from the data in figure 4.

Sustainable increases in aGalA and VCN

They claim many-fold increases in enzyme activities in leukocytes and increased vector copy numbers that are sustained over time. Indeed at day 180 there is a clear improvement (figure 1), however, figure 2 shows a decline in both. The table one in the rebuttal shows that leukocyte aGalA is still manyfold increased, which is convincing. However, in light of this, I find the clinical and biochemical responses disappointing, which, again, suggest that the gene therapy may fail to target the organs that need to be supplemented the most: the heart and kidneys.

Antibodies:

The results are promising with respect to disappearance of antibodies. As with pegunigalsidase (which has a long circulation time), it is possible that the ab titers also are reduced because of complex formation with circulating aGalA. However, so far, with the decline in activity and VCN, there is no sign of reappearance of abs.

Eligibility criteria

I would suggest that in addition to the criteria to stop ERT, criteria for failure/progression of disease are defined (which could be a signal to restart ERT or a signal that the patient has passed a critical threshold for reversibility)

Final comment:

Despite all the criticisms I have forwarded, I do believe that this is important work and should be published. In the rebuttal, the authors emphasize several times that this is early work and further studies are needed. In fact, they say that " Our gene therapy study was a pilot safety study." I suggest to change the manuscript in this way, showing that this approach is safe, but that effectiveness is still to be awaited and that there may be concerns on targeting of the affected organs. These innovative approaches should be supported, improved and in the end hopefully lead to a more effective therapy than what we currently have available.

Reviewer #5:

Remarks to the Author:

The reporting of the clinical trial could be improved.

In the current version of the paper, it is not clear what the proposed trial design is; how each patient would be recruited in this Phase I trial. Also, the authors have not stated explicitly what their primary and secondary objectives are, as well as their primary and secondary outcomes are. At the minimum, they should state what their primary and key secondary objectives/outcomes are. Those information can be found in the protocol but they are not available in this manuscript.

In the protocol, their proposed sample size is 6. It would be useful for the authors to explain why only 5 patients were reported here. What was the reason for not recruiting to 6 patients?

As safety is the primary objective here, the authors should report the AEs/SAEs for each patient that are used for decision-making to continue/stop recruitment. Supplementary Table 2 does not provide the AEs at a patient level.

As stated in the protocol:

"After each patient completes the 1 month post-transplant follow-up visit, a safety review meeting will take place with the Clinical Trial Steering Committee (CTSC) (refer to section 6.1.2) and Ozmosis Research Inc. to review the safety data (refer to section 6.2). If there is no protocol treatment-related Serious Adverse Events (any SAEs including failure to engraft deemed by CTSC to be definitely, probably or possibly related to infusion of transduced autologous CD34+ cells), or any significant safety concerns as defined in section 6.1.1, a decision will be made based on the data submitted for the patient regarding whether the study will be re-opened for the next patient to start treatment phase of the study. Only when it is deemed safe can the next patient start the treatment phase of the study."

Please provide sufficient information to help the reader understand the decision-making process at

the safety review meetings throughout the trial to improve transparency and reproducibility.

Minor points:

Pg 3 Sub-heading: It would be more appropriate to state "functional efficacy assessments" rather than "efficacy assessments"

Under Study Design, they should explain how the sample size was determined.

Under Results:

Please state dates defining the periods of recruitment and follow-up

REVIEWER COMMENTS

Reviewer #1 (Remarks to the Author):

The authors have clarified most of my concerns. Nevertheless, in opinion of this reviewer some information related to insertional site analyses would be necessary to verify important safety aspects of this first Phase I gene therapy trial in Fabry patients.

We accept the Reviewer's point and agree that these analyses are of importance. We have received permission from all the involved IRBs to do this study. We have also contracted with the group in Hannover to do LAM-PCR analyses. However, given all the institutional delays and reduced patient visits that are happening due to COVID-19, we do not expect to have this additional data until sometime in 2021. Given the competitive nature of the Phase 1 Fabry gene therapy arena with at least two competing ongoing studies, it is vital that our report is published prior to receipt of these data.

I still think that statements indicating that this gene therapy protocol results in sustained correction responses need to be tempered, since in several instances this is not evident.

We take the Reviewer's point. The manuscript has been modified in several places to temper these statements and to focus on the fact that our primary endpoint in this study was safety.

Additionally, I strongly recommend to include some discussion about possibilities for improving this approach of gene therapy in Fabry disease. For example, a paragraph mentioning advantages and limitations of giving a more myelo-suppressive conditioning would be useful. Similarly, the authors mention that a low number of VCN/cell is an advantage, although this reviewer considers that it would be convenient to increase this VCN to facilitate a higher release of the enzyme, and thus to achieve a higher therapeutic efficacy.

We take the Reviewer's points. We have now added such a paragraph to the Discussion on page 10, as indicated below:

"It is possible that a more myelo-suppressive conditioning regimen would result in better engraftment of LV/AGA transduced cells - though this hypothesis remains to be tested. In selection of a partial myelo-

Jeffrey A. Medin, PhD

MACC Fund Professor, Departments of Pediatrics and Biochemistry
Vice Chair of Research Innovation, Department of Pediatrics
Research Director, Division of Pediatric Hematology/Oncology/BMT
Director GMP Vector Lab

ablative conditioning regimen, we considered that our prospective patients were relatively healthy males with Fabry disease who were receiving ET prior to recruitment to our trial. Unlike most other acute disorders that have been treated by gene therapy, Fabry disease is a chronic disease. As such, our clinical team and our regulatory body were reluctant to implement a full myelo-ablative conditioning regimen. Such protocols are more invasive, are accompanied by greater number of adverse events, and have the potential for future oncogenic events. Likewise, infection of our target cells at a higher MOI or multiple infection courses may also lead to higher transgene expression; however, GMP-grade, recombinant LV was already limiting in our trial."

Reviewer #2 (Remarks to the Author):

In this report, Kahn et al. present the results of a Phase 1, pilot study of lentiviral-mediated gene therapy for Fabry disease. The investigators enrolled and treated 5 subjects and were able to obtain successful transduction of CD34+ hematopoietic progenitors that were infused after reduced intensity conditioning with melphalan. The procedure resulted in the appearance of gene-corrected peripheral blood cells and the intracellular and plasma production of alpha-Gal A activity in all patients, which demonstrated the feasibility of the approach. These results are encouraging and support further development and application of the procedure.

Major comments

In the abstract and on pages 9-10, the sentences "all patients demonstrated a sustained corrective response" and " All patients had... a reduction in plasma and urine Gb₃ levels, and plasma and urine lyso-Gb₃ levels" appear to be over-interpretations of the results.

We take the Reviewer's point. The manuscript has been changed in several places to temper those statements and to focus on the fact that our primary endpoint in this study was safety.

It would seem that "corrective" effects should be only considered for patients 1, 3, and 4 who stopped ET. While it is true that treatment led to increased a-gal A activity in all 5 subjects, Pt. 1 showed an increase of urine Gb₃ and lyso-Gb₃ after ET stop, Pt. 3 showed levels of plasma and urine Gb₃ at the last determination that appear to be higher than pre-gene therapy baseline values, and Pt. 4 showed an increase of plasma and urine Gb₃ and lyso-Gb₃ after ET stop. In addition, effects of the treatment on the clinical parameters shown in Figure 4 are not clearly evident. Based on these observations, it seems

Jeffrey A. Medin, PhD

MACC Fund Professor, Departments of Pediatrics and Biochemistry
Vice Chair of Research Innovation, Department of Pediatrics
Research Director, Division of Pediatric Hematology/Oncology/BMT
Director GMP Vector Lab

warranted that conclusions regarding any corrective effects that may have been observed be significantly tempered.

As above - we take the Reviewer's point. The manuscript has been changed in several places to temper those statements and to focus on the fact that our primary endpoint in this study was safety.

Minor issues

Page 5: the Authors should consider rephrasing the sentence "Gene therapy enables Fabry patients to receive ..." to indicate that, if successful, gene therapy may result in beneficial results with a single treatment.

We take the Reviewer's point. We have changed the manuscript on page 11. The new sentence reads:

"If successful, LV-mediated gene therapy may result in beneficial outcomes for Fabry patients with a single treatment. "

Page 5: the sentence "A single ex-vivo LV infection..." should be changed to " A single ex-vivo LV transduction..."

This change has been made as requested by the Reviewer.

Page 6: was ET stopped before treatment to provide selective advantage to gene-corrected cells?

The reason ET was stopped here in this gene therapy trial is actually much simpler: ET was stopped primarily so we could obtain a consistent cohort of baseline measurements. The following statement has been added to p6 in the Results section:

"ET was stopped so that we could obtain a consistent cohort of baseline measurements."

However, there was also a regulatory necessity for this ET cessation: In Canada, patients cannot be on two experimental therapies at once. REPLAGAL was not approved at the time of initiation of this clinical trial; it was considered experimental at that time. At least 30 days was required to switch any REPLAGAL patients that might be recruited to our trial to FABRAZYME.

Jeffrey A. Medin, PhD

MACC Fund Professor, Departments of Pediatrics and Biochemistry
Vice Chair of Research Innovation, Department of Pediatrics
Research Director, Division of Pediatric Hematology/Oncology/BMT
Director GMP Vector Lab

Figure 2a: a tentative explanation for the progressive reduction of alpha-gal A activity and VCN should be added to the Discussion, especially addressing the possibility of toxicity of high levels of alpha-gal A expression or high LV copy numbers.

To address the Reviewer's first point here we have modified the following statement to the text on page 9.

"Values declined over time, possibly due to an exhaustion of transduced short-term repopulating HSPCs, but remained above what is typically observed in Fabry patients and well above pre-treatment levels in all 5 cases."

As to the Reviewer's other points: we have already addressed the next one concerning possible transgene product toxicity in the text of the manuscript itself. In the Introduction, page 5, we have the following statement:

"Transgenic mice, with tissue α -gal A activity >10,000 times endogenous levels, were healthy and did not have altered cyto-architecture^{18,19}; thus high levels of α -gal A may not be deleterious."

To address the Reviewer's last point, we remind the Reviewer that we were aiming for a relatively low VCN in this pilot gene therapy trial, which was successfully achieved. Furthermore, the Reviewer may be familiar with other trials in the field, also involving LVs and HSPCs wherein higher VCNs were obtained:

-Ferrura *et al.* Lentiviral haemopoietic stem/progenitor cell gene therapy for treatment of Wiskott-Aldrich syndrome: interim results of a non-randomised, open-label, phase 1/2 clinical study. *Lancet Haematol* 6(5):e239, 2019. Median VCN of 2.4. These cells were stable; no clonal expansion or leukemia has been recorded to date in that study.

-Biffi *et al.* Lentiviral hematopoietic stem cell gene therapy benefits metachromatic leukodystrophy. *Science* 341:1233158, 2013. VCN was 2.5 to 4.4 following transductions. BM VCN was 0.9 to 1.9 out to 25 months. Polyclonal reconstitution of hematopoiesis was seen with no evidence of vector genotoxicity.

-Aiuti *et al.* Lentiviral hematopoietic stem cell gene therapy in patients with Wiskott-Aldrich Syndrome. *Science* 341:1233151, 2013. VCN was 2.3 +/- 0.6 following transduction. PB VCNs ranged from 0.6 to 3.0 depending on the lineage. Multilineage and polyclonal hematopoietic cell engraftment was seen.

Jeffrey A. Medin, PhD

MACC Fund Professor, Departments of Pediatrics and Biochemistry
Vice Chair of Research Innovation, Department of Pediatrics
Research Director, Division of Pediatric Hematology/Oncology/BMT
Director GMP Vector Lab

-Kohn *et al.* Lentiviral gene therapy for X-linked chronic granulomatous disease. *Nat Med* 26:200, 2020. VCN here ranged from 0.7 to 5.5 in transduced CD34+ cells. VCN ranged from 0.4 to 1.8 in peripheral blood neutrophils. No clonal dysregulation was seen.

As well, this last comment is also somewhat incongruent with the comments of Reviewer 1.

Page 7: please clarify the meaning of "(i.e. prior to mobilization)".

This was an error during editing of the manuscript. This modifying comment has been removed.

Figure 5: the reference arrows are missing.

The reference arrows have been added to this figure.

Methods, page 11: please specify what was considered a "low" blood CD34+ count.

We thank the Reviewer for pointing out this omission. We have rewritten that sentence to specify a "low" count as requested. The new sentence on page 12 under Study Design reads as follows:

If peripheral blood CD34⁺ counts were low (i.e. if the predicted CD34⁺ count was <50% of our target of 12.5 x 10⁶/kg), filgrastim was supplemented with plerixafor (240 µg/kg).

Methods: a brief description of the transduction conditions (schedule, growth factors, MOI) would be of interest.

We thank the Reviewer for pointing out this omission. We have added the following sentence to the Methods section on page 13.

"The transduction protocol was described in detail.²⁰

Jeffrey A. Medin, PhD

MACC Fund Professor, Departments of Pediatrics and Biochemistry
Vice Chair of Research Innovation, Department of Pediatrics
Research Director, Division of Pediatric Hematology/Oncology/BMT
Director GMP Vector Lab

Reviewer #3 (Remarks to the Author):

Lentivirus-Mediated Gene Therapy for Fabry Disease

The authors have revised their original manuscript, which is improved in several ways. However, there are still some major issues.

Cross-correction

Critical for gene-therapy in Fabry disease using this approach is the possibility that cross corrections occurs, since the primary storage is not in the hematopoietic system. The authors point out that in the studies by Medin and Huang, cross correction was shown. However, the construct used by Medin et al resulted in an enzyme that could be taken up in a mannose-6-phosphate dependent way: the same as the recombinant enzymes used for ERT.

We point out to the Reviewer that the vector construct used in the Huang et al. (2017) paper was the identical construct used in our present clinical trial. Indeed, the LVs we have been using since our first publication on the use of that vector system for gene therapy of Fabry disease (Yoshimitsu *et al.* Bioluminescent imaging of a marking transgene and correction of Fabry mice by neonatal injection of recombinant lentiviral vectors. *Proc Natl Acad Sci U S A.* 101: 16909-14, 2004) have all coded for the wild-type human form of α -gal A. Further, even our earliest gene therapy papers using recombinant gamma retroviral vectors encoded for the human form of α -gal A as well. We are happy to walk the Reviewer through the molecular biology of this: even when the vector backbone or promoter or bicistronic transgene differed, the same α -gal A transcript was made. This was translated into the same primary amino acid sequence encoding for human α -gal A. We also wish to point out that we have demonstrated this cross-correction in Fabry disease in numerous additional independent publications including:

Medin JA *et al.* Correction in trans for Fabry disease: expression, secretion and uptake of alpha-galactosidase A in patient-derived cells driven by a high-titer recombinant retroviral vector. *Proc Natl Acad Sci U S A.* 93: 7917-22, 1996.

Takenaka T *et al.* Enzymatic and functional correction along with long-term enzyme secretion from transduced bone marrow hematopoietic stem/progenitor and stromal cells derived from patients with Fabry disease. *Exp Hematol.* 27: 1149-59, 1999.

Jeffrey A. Medin, PhD

MACC Fund Professor, Departments of Pediatrics and Biochemistry
Vice Chair of Research Innovation, Department of Pediatrics
Research Director, Division of Pediatric Hematology/Oncology/BMT
Director GMP Vector Lab

Takenaka T *et al.* Circulating alpha-galactosidase A derived from transduced bone marrow cells: relevance for corrective gene transfer for Fabry disease. *Hum Gene Ther.* 10: 1931-9, 1999.

Takenaka T *et al.* Long-term enzyme correction and lipid reduction in multiple organs of primary and secondary transplanted Fabry mice receiving transduced bone marrow cells. *Proc Natl Acad Sci U S A.* 97: 7515-20, 2000.

Yoshimitsu M *et al.* Bioluminescent imaging of a marking transgene and correction of Fabry mice by neonatal injection of recombinant lentiviral vectors. *Proc Natl Acad Sci U S A.* 101: 16909-14, 2004.

Yoshimitsu M *et al.* Efficient correction of Fabry mice and patient cells mediated by lentiviral transduction of hematopoietic stem/progenitor cells. *Gene Ther.* 14: 256-65, 2007.

Liang SB *et al.* Multiple reduced-intensity conditioning regimens facilitate correction of Fabry mice after transplantation of transduced cells. *Mol Ther.* 15: 618-27, 2007.

Pacienza N *et al.* Lentivector transduction improves outcomes over transplantation of human HSCs alone in NOD/SCID/Fabry mice. *Mol Ther.* 20: 1454-61, 2012.

Provençal P *et al.* Relative distribution of Gb₃ isoforms/analogs in NOD/SCID/Fabry mice tissues determined by tandem mass spectrometry. *Bioanalysis.* 8: 1793-807, 2016.

Huang J *et al.* Lentivector Iterations and Pre-Clinical Scale-Up/Toxicity Testing: Targeting Mobilized CD34⁺ Cells for Correction of Fabry Disease. *Mol Ther Methods Clin Dev.* 5: 241-258, 2017.

However as elegantly discussed by Beck and Cox in a recent paper, (Mol Genet Metab Rep. 2019 Dec; 21: 100529.) the molecular human form of α -galactosidase in plasma is the mature 46 kDa enzyme and not the high-uptake, mannose 6-phosphorylated form.

Answering this comment is beyond the scope of this present manuscript submission. Here we are reporting the safety results of a pilot study of gene therapy for Fabry disease. As well, it is unfortunate that the elegant paper by Beck and Cox Comment: Why are females with Fabry disease affected? *Mol Genet Metab Rep* 21:100529, 2019, chose only to focus on results from a single manuscript (Fuller *et al.* Absence of α -galactosidase cross correction in Fabry heterozygote skin fibroblasts. *Mol Genet*

Jeffrey A. Medin, PhD

MACC Fund Professor, Departments of Pediatrics and Biochemistry
Vice Chair of Research Innovation, Department of Pediatrics
Research Director, Division of Pediatric Hematology/Oncology/BMT
Director GMP Vector Lab

Metabol 114: 268-273, 2015) in their commentary, and one wherein α -gal A was NOT overexpressed, thereby overlooking all of the data (for example, a rich history of LV and AAV mediated transduction of α -gal A in numerous studies) that have demonstrated functional uptake and cross-correction results even in 'difficult' tissues for Fabry.

In the Huang paper, it is not clear how the enzyme is taken up and whether there is (sufficient) man-6-phosphorylation. Indeed, an increase in a-GalA activity in PBMC's and plasma is shown (since the hematopoietic system is transduced), but hardly in tissues: only in organs that contain many hematopoietic cell line derived cells such as macrophages (spleen and liver) but not in heart and kidney, the crucial organs to target in Fabry disease. What is more, no significant Gb₃ clearance was shown in these organs.

Please see comments above. Cross correction in Fabry disease gene therapy has been demonstrated by the Medin group in multiple publications, even in difficult to access tissues - such as the brain. Functional uptake and cross correction have also been validated by Gb₃ changes; not just enzyme accumulation. The AAV community has also demonstrated similar results in several publications. This Phase 1 clinical trial was a safety trial devoid of invasive biopsies; consequently, our analysis of Gb₃ and lyso-Gb₃ levels was restricted to plasma and urine.

Biochemistry

Another major issue is the response in terms of biochemical parameters: plasma and urine globotriaosylceramide (Gb₃) and globotriaosylsphingosine (lyso-Gb₃). In the abstract, reductions in plasma and urine Gb₃ and lyso-Gb₃ are mentioned as the most robust outcome measures.

We agree with the Reviewer's (and Editor's) point. Safety is our primary focus in this study; this aspect has been brought forward in the Abstract and the other, overly enthusiastic, efficacy comments have been tempered.

However, the data show different and perhaps even non-sustainable results. Surprisingly, in their rebuttal they argue that there is no clinical correlate with these biochemical parameters, and their argument to use them is "to keep with current literature". This is not a valid argument. Either you are convinced that these parameters are of value, which I believe is true (in the sense that successful enzyme therapy in classical males is always followed by a decline in Gb₃ and lyso-Gb₃, which is reversed with dose interruptions or antibody formation), or you do not believe that this is a valid parameters, but then you should not use it as the most important clinical endpoint.

Jeffrey A. Medin, PhD

MACC Fund Professor, Departments of Pediatrics and Biochemistry
Vice Chair of Research Innovation, Department of Pediatrics
Research Director, Division of Pediatric Hematology/Oncology/BMT
Director GMP Vector Lab

We take the Reviewer's comments and have tempered the manuscript accordingly. As mentioned above, safety is now the most important clinical endpoint. Notably, we have shown that our trial was safe. We are now reporting the other data we have accumulated as 'matter-of-factly' as possible as these parameters would be expected to be monitored in such a trial involving Fabry patients.

Clinical outcomes

Following criticism on the course of the biochemical parameters and lack of clinical data, the authors have added new data (weight, eGFR, urinary protein, troponin, and LVMI). Unfortunately, these data do not support the conclusion that gene therapy is effective. They state that weight is increasing (which is obviously a parameters that can be influenced by many factors), however, there is hardly any change in weight, with the exception of one obese patient adding more weight (nr 5). Most patients appear to have very limited organ damage. That proteinuria and eGFR are relatively stable in those with preserved kidney function is not exceptional. In patient 2, with renal impairment, there is a gradual further decrease in eGFR, consistent with observations during ERT. The other parameters are all over the place, due to variations in measurements, I assume, which is frequently observed in these patients. Hence, no support for clinical benefit of gene therapy can be concluded from the data in figure 4.

We believe it is important to show this clinical data for this first-in-the-world gene therapy study for Fabry disease. We have decreased all the overly enthusiastic statements in the manuscript that had discussed efficacy.

Sustainable increases in a-GalA and VCN

They claim many-fold increases in enzyme activities in leukocytes and increased vector copy numbers that are sustained over time. Indeed, at day 180 there is a clear improvement (figure 1), however, figure 2 shows a decline in both. The table one in the rebuttal shows that leukocyte a-GalA is still manyfold increased, which is convincing. However, in light of this, I find the clinical and biochemical responses disappointing, which, again, suggest that the gene therapy may fail to target the organs that need to be supplemented the most: the heart and kidneys.

There is a definitely a decline in both leukocyte and plasma α -gal A activity, nonetheless some level is still maintained as the Reviewer notes. According to Canadian assessment criteria, these patients do not now have Fabry disease. Beyond this - there is nothing to respond to in this comment as we do not have additional heart and kidney tissue biochemical data. We were not powered for this. It is also possible

Jeffrey A. Medin, PhD

MACC Fund Professor, Departments of Pediatrics and Biochemistry
Vice Chair of Research Innovation, Department of Pediatrics
Research Director, Division of Pediatric Hematology/Oncology/BMT
Director GMP Vector Lab

that the Reviewer may not understand the difficulties of clinical assessments in Fabry disease. Longer term follow-ups will be required to assess whether heart and kidney function is affected. Patient 1 already was displaying Fabry cardiac symptoms at recruitment, while Patient 2 had Fabry kidney symptoms prior to our trial.

Antibodies:

The results are promising with respect to disappearance of antibodies. As with pegunigalsidase (which has a long circulation time), it is possible that the ab titers also are reduced because of complex formation with circulating a-GalA. However, so far, with the decline in activity and VCN, there is no sign of reappearance of abs.

We agree with the Reviewer that the lack of antibody responses in all patients is very encouraging for the long-term prospects of this approach.

Eligibility criteria

I would suggest that in addition to the criteria to stop ERT, criteria for failure/progression of disease are defined (which could be a signal to restart ERT or a signal that the patient has passed a critical threshold for reversibility)

We thank the Reviewer for this suggestion. However, Health Canada has not requested delineation of the failure of ET cessation or progression of disease. It would be highly unusual for this to be described in the manuscript and not stipulated by the regulator. We would like to remind the Reviewer that our group has monthly meetings to discuss clinical and biochemical parameters. Each patient is also evaluated by their own treating physician. Further, the Clinical Trial Steering Committee of the FACTs Team (headed by Dr. Foley) has been actively monitoring all aspects of patient safety throughout the trial. Lastly, Dr. West leads the Canadian Fabry Disease Initiative, and has access to long-term data for all Fabry patients in this study.

Final comment:

Despite all the criticisms I have forwarded, I do believe that this is important work and should be published.

We are encouraged by the enthusiasm of this Reviewer regarding our submission. We trust that the modifications made to this version of the manuscript will enable timely publishing of this 'first-in-class' study.

MACC Fund Professor, Departments of Pediatrics and Biochemistry
Vice Chair of Research Innovation, Department of Pediatrics
Research Director, Division of Pediatric Hematology/Oncology/BMT
Director GMP Vector Lab

In the rebuttal, the authors emphasize several times that this is early work and further studies are needed. In fact, they say that " Our gene therapy study was a pilot safety study." I suggest to change the manuscript in this way, showing that this approach is safe, but that effectiveness is still to be awaited and that there may be concerns on targeting of the affected organs. These innovative approaches should be supported, improved and in the end hopefully lead to a more effective therapy than what we currently have available.

We take the Reviewer's point. We have made the changes requested to the manuscript.

Reviewer #5 (Remarks to the Author):

The reporting of the clinical trial could be improved.

In the current version of the paper, it is not clear what the proposed trial design is

We thank the Reviewer for identifying this omission. The trial design (pilot, single-arm) and primary endpoints (safety and toxicity) have been added to the Abstract.

how each patient would be recruited in this Phase I trial.

We have added the following statement to the beginning of the Results section:

"Male patients with known Fabry disease from the Canadian Fabry Disease Initiative study of ET were approached for possible enrollment".

Also, the authors have not stated explicitly what their primary and secondary objectives are, as well as their primary and secondary outcomes are. At the minimum, they should state what their primary and key secondary objectives/outcomes are. Those information can be found in the protocol but they are not available in this manuscript.

We thank the Reviewer for this comment. We have added the following paragraph at the beginning of the Results section on page 6.

"The primary objectives of this study were to determine the safety and toxicity of autologous stem cell transplantation with mobilized CD34+ hematopoietic cells transduced with a lentiviral vector containing human codon-optimized α -gal A in adult male Fabry disease patients. Several secondary objectives were

Jeffrey A. Medin, PhD

MACC Fund Professor, Departments of Pediatrics and Biochemistry
Vice Chair of Research Innovation, Department of Pediatrics
Research Director, Division of Pediatric Hematology/Oncology/BMT
Director GMP Vector Lab

analyzed including monitoring levels of α -gal A activity in plasma, peripheral blood leukocytes and bone marrow-derived mononuclear cells; measuring levels of Gb₃ and lyso-Gb₃ in plasma and urine. In addition, the transduction efficiency of CD34+ hematopoietic cells was assessed as well as the presence and persistence of marked cells expressing α -gal A in peripheral blood. The primary endpoint of this study was toxicity as assessed using the National Cancer Institute of Canada (NCIC) Common Terminology Criteria for Adverse Events (CTCAE), Version 4.03.”

In the protocol, their proposed sample size is 6. It would be useful for the authors to explain why only 5 patients were reported here. What was the reason for not recruiting to 6 patients?

The sample size was determined exclusively by the amount of LV available to complete the trial. We initially planned on treating six patients, however, there was only adequate vector available for treating five patients. This is described in the “*Lentiviral Vector*” section of the Methods.

As safety is the primary objective here, the authors should report the AEs/SAEs for each patient that are used for decision-making to continue/stop recruitment. Supplementary Table 2 does not provide the AEs at a patient level.

We have added the information as requested by the Reviewer to illustrate which patients displayed specific adverse events.

As stated in the protocol:

“After each patient completes the 1 month post-transplant follow-up visit, a safety review meeting will take place with the Clinical Trial Steering Committee (CTSC) (refer to section 6.1.2) and Ozmosis Research Inc. to review the safety data (refer to section 6.2). If there is no protocol treatment-related Serious Adverse Events (any SAEs including failure to engraft deemed by CTSC to be definitely, probably or possibly related to infusion of transduced autologous CD34+ cells), or any significant safety concerns as defined in section 6.1.1, a decision will be made based on the data submitted for the patient regarding whether the study will be re-opened for the next patient to start treatment phase of the study. Only when it is deemed safe can the next patient start the treatment phase of the study.”

Please provide sufficient information to help the reader understand the decision-making process at the safety review meetings throughout the trial to improve transparency and reproducibility.

Jeffrey A. Medin, PhD

MACC Fund Professor, Departments of Pediatrics and Biochemistry
Vice Chair of Research Innovation, Department of Pediatrics
Research Director, Division of Pediatric Hematology/Oncology/BMT
Director GMP Vector Lab

We have amended the text to provide greater clarity. The following statement was added on page 14

“Safety review meetings were conducted by the CTSC one month after the transplant date for each patient. The next patient was then eligible for recruitment to the study only after the CTSC reviewed all of the safety data and unanimously concluded that there has been no treatment-related serious adverse events.”

Minor points:

Pg 3 Sub-heading: It would be more appropriate to state “functional efficacy assessments” rather than “efficacy assessments”

We take the Reviewer's point; this change has been made to the text on page 13.

Under Study Design, they should explain how the sample size was determined.

As mentioned above - the sample size was determined exclusively by the amount of LV available to complete the trial. We initially planned on treating six patients, however, there was only adequate vector available for treating five patients. This is described in the “*Lentiviral Vector*” section of Methods.

Under Results:

Please state dates defining the periods of recruitment and follow-up

We have added the periods of recruitment from September 2016 to October 2018 to the text. The duration of follow-up for the data reported in this manuscript extended from January 2017 to February 2020. Ongoing follow-up of the patients recruited to this trial will extend to February 2024

Jeffrey A. Medin, PhD

MACC Fund Professor, Departments of Pediatrics and Biochemistry
Vice Chair of Research Innovation, Department of Pediatrics
Research Director, Division of Pediatric Hematology/Oncology/BMT
Director GMP Vector Lab

Reviewers' Comments:

Reviewer #3:

Remarks to the Author:

The second revision of the manuscript addresses most of my concerns. The authors have now appropriately changed the focus of the paper towards safety and tempered their enthusiasm on efficacy.

So, my last comments are to make a couple of small changes in order to further emphasize this

- abstract. Delete the sentence: "All patients are eligible to discontinue enzyme therapy due to sustained production of intracellular and secreted a-gal A;"

Change to: "Three patients have elected to discontinue enzyme therapy."

Introduction:

"Gene therapy enables Fabry patients" change to "Gene therapy in theory would enable Fabry patients"

Discussion: "The enzyme values may also be reaching an asymptote reflecting engraftment of transduced long-term HSPCs." change to: "Whether the enzyme values reach an asymptote reflecting engraftment of transduced long-term HSPCs remains to be determined."

Reviewer #6:

Remarks to the Author:

This manuscript reports results from a pilot, single-arm safety trial of five patients with Type 1 Fabry disease who were treated with lentivirus-mediated gene therapy targeting CD34+ hematopoietic stem/progenitor cells engineered to express alpha-galactosidase A. This novel treatment may improve outcomes without traditional enzyme therapy. The authors reported safety and outcomes for this pilot study. This manuscript appears to present important progress in development of a novel treatment for Fabry disease and reports important and complete data from their work.

The authors have satisfactorily addressed all of the comments made by the original Reviewer #5 including better describing the:

1. Study design
2. Recruitment process
3. Primary and secondary objectives
4. Primary endpoint
5. Why the sample size was n=5 whereas the protocol targeted n=6 (not explicitly discussed, but authors note twice that the amount of lentiviral vector was the limiting factor resulting in sample size of 5)
6. AEs/SAEs by subject (Supp. Table 2)
7. Safety and enrollment decision-making process
8. Wording for "Functional efficacy assessments"
9. Dates defining recruitment and follow-up

Minor comments:

1. In the abstract the authors state, "...the first patient is now out more than three years." This is reiterated on p. 9 (first sentence of Discussion). However, on page 10 in the last third of the second paragraph of the Discussion, they state, "...LVMI were stable for all patients throughout the study period (ranging from 12-33 months post-infusion)..." This is inconsistent with the first two statements of time elapsed and should be updated.

2. Page 12: section called "Patients": the penultimate sentence is redundant with the second

sentence of this paragraph (both state that patients provided written informed consent).

REVIEWERS' COMMENTS

Reviewer #3 (Remarks to the Author):

The second revision of the manuscript addresses most of my concerns. The authors have now appropriately changed the focus of the paper towards safety and tempered their enthusiasm on efficacy.

So, my last comments are to make a couple of small changes in order to further emphasize this

- abstract. Delete the sentence: "All patients are eligible to discontinue enzyme therapy due to sustained production of intracellular and secreted a-gal A;"

Change to: "Three patients have elected to discontinue enzyme therapy."

We have made this change verbatim in the Abstract.

Introduction:

"Gene therapy enables Fabry patients" change to "Gene therapy in theory would enable Fabry patients"

We have made this change verbatim in the Introduction (except with commas to separate out the conjunctive adverb):

Gene therapy, in theory, would enable Fabry patients to receive a single treatment that could be more effective than current options and free them from ET.

Discussion: "The enzyme values may also be reaching an asymptote reflecting engraftment of transduced long-term HSPCs." change to: "Whether the enzyme values reach an asymptote reflecting engraftment of transduced long-term HSPCs remains to be determined."

We have made this change verbatim in the Discussion.

Reviewer #6 (Remarks to the Author):

This manuscript reports results from a pilot, single-arm safety trial of five patients with Type 1 Fabry disease who were treated with lentivirus-mediated gene therapy targeting CD34+ hematopoietic stem/progenitor cells engineered to express alpha-galactosidase A. This novel treatment may improve outcomes without traditional enzyme therapy. The authors reported safety and outcomes for this pilot

study. This manuscript appears to present important progress in development of a novel treatment for Fabry disease and reports important and complete data from their work.

We thank the Reviewer for these comments.

The authors have satisfactorily addressed all of the comments made by the original Reviewer #5 including better describing the:

1. Study design
2. Recruitment process
3. Primary and secondary objectives
4. Primary endpoint
5. Why the sample size was n=5 whereas the protocol targeted n=6 (not explicitly discussed, but authors note twice that the amount of lentiviral vector was the limiting factor resulting in sample size of 5)
6. AEs/SAEs by subject (Supp. Table 2)
7. Safety and enrollment decision-making process
8. Wording for "Functional efficacy assessments"
9. Dates defining recruitment and follow-up

We thank the Reviewer for these comments.

Minor comments:

1. In the abstract the authors state, "...the first patient is now out more than three years." This is reiterated on p. 9 (first sentence of Discussion). However, on page 10 in the last third of the second paragraph of the Discussion, they state, "...LVMI were stable for all patients throughout the study period (ranging from 12-33 months post-infusion)..." This is inconsistent with the first two statements of time elapsed and should be updated.

We thank the Reviewer for pointing out this inconsistency. To align the statements of time elapsed with the study period statement, we have changed the text in the Abstract and the first part of the Discussion to 'nearly 3 years'.

2. Page 12: section called "Patients": the penultimate sentence is redundant with the second sentence of this paragraph (both state that patients provided written informed consent).

We thank the Reviewer for catching this overlap. We have removed the second iteration of this statement.